# Tradition-Driven Religiosity on the Internet

**Márta Katalin Korpics** [1,*], **István József Béres** [2] **and Anna Veronika Hommer** [2]

1. Department of Public Management and Information Technology, Ludovika University of Public Service, 1083 Budapest, Hungary
2. Department of Liberal Arts, Károli Gáspár University of Reformed Church in Hungary, 1091 Budapest, Hungary; beres.istvan.jozsef@kre.hu (I.J.B.); hommer.anna@kre.hu (A.V.H.)
* Correspondence: korpics.marta.katalin@uni-nke.hu

**Abstract:** The evolution of the media environment and the expanding use of social media pose a question about how the public sphere has changed in the last decade and what standards churches and religious communities can refer to in order to thrive and be present. It is vital to see that the nature of mass communication is different from the way religious ideas are conveyed. Consequently, there is a constant threat that, if mass media report on religions and religious activities, even with the greatest possible neutrality, they can easily falsify them. This paper sets out to explore this paradox; we examine the social media activity of a tradition-driven religious community. This research focuses on particular phenomena that lead to general assumptions. Despite the fact that the online reality is not the primary space of the researched community, its activity has moved towards cyber space because of the diminishing presence of interpersonal and group relationships. This paper analyses this community's Facebook presence by applying quantitative and qualitative methods.

**Keywords:** reflexive modern religiosity; mediatisation; social media; social media consumption; tradition driven





## 1. Introduction

Religious life is colourful; it can be experienced outside of denomination (for example, by people who claim to be religious in their own way), but it can also be experienced within a church (by those who claim to belong to one). Every church is a unique organisation characterised by different activities that manifest themselves both within and outside it. These activities can be called patterns (Davie 2000), which are what each religion is made up of, with each church representing that religion, denomination, and community. The wide variety of patterns is exciting and, if we look at them from a social science perspective, we can see they show the significance of a religious group or a church community. As social scientists, we are aware that the patterns we can see and study cannot provide us with the full picture of the religiosity of either an individual or an organisation; nonetheless, they do refer to the full reality of a religion and represent a church. In this way, it is important to analyse them, as they allow for a better understanding of the religion or church (denomination) in question. We can also better assess what the activities of these various organisational units mean for society at a given moment in time.

Sociology of religion collects and analyses data on religion, religiosity, religious communities, churches, and ecclesial communities. There are other specific examinations as well, provided by anthropology of religion. As a new discipline, communication research is related to this field because, on the one hand, it studies the connections between religion and media and, on the other hand, it analyses the external and internal communication of religious communities (Andok 2008, 2022; Andok et al. 2023; Béres 2022; Hommer 2021, 2022; Korpics 2020; Rončáková 2008). Communities change and progress socially, which is why our research requires the inclusion of new perspectives. There is no consensus on what defines a community (Hillary 1955); however, there is consensus on the fact that

communication is relevant in every approach to community. For a community to be formed, there needs to be internal communication, shared experiences, and solidarity among the members and, also, external communication that connects them to society. A community is bound to have traditions as well, which are kept alive and passed on through communication. The terms community and communication have received attention from researchers, and several scholars have connected the two terms: Rosengren's definition of community is based on communication (Rosengren 2000); for Depew and Peters, communication is what holds a community together (Depew and Peters 2000); Carey suggests that a new communication model would be appropriate for studying changing social circumstances because, he posits, communication is the representation of shared convictions, which is a symbolic process that creates and sustains reality (Carey 1992).

Even though this paper focuses on particular phenomena, these relate to important questions concerning the religiosity of our era. One of the greatest challenges for traditional churches is finding a way to convey religious messages in a space that has changed and continues to change into what are or should be the new spaces of religious public life (Pope Francis 2019; Spadaro 2014). The Christian tradition has been gradually losing its dominance in society since before the modern era. Similarly, its characteristic of being personally obligatory—an internalised, lived, and confessional faith—has also lost its dominance, even for those believers who are still culturally Christian. The most recent schism in how traditions are passed began in the 1960s and 1970s. The virtual space that started to build up from then provided room for real-time, dynamic, impulsive communication, instead of the slow, local transmission of tradition that existed previously. This space is driven by in-the-moment, current communication to such an extent that almost any other form of information, such as normative knowledge or tradition, suffers a disadvantage. This space, which is updated moment by moment, excludes the validity of anything old and traditional. Here, opinions are exchanged in a constant flow, and texts, photos, memes, emojis, short videos, and likes, for better or worse, can have an impact and are able to generate an immediate response. This dense multitasking in a world full of multimedia is resulting in a growing distance from any kind of tradition. This process is not against religion; its goal is not the destruction of religion; yet, every traditional institution that is not built on volatility and optionality gets in its way. This is exceptionally tragic for churches since they rely heavily on tradition; what is more, the basic element of their function is relaying messages (Moingt 2010; Luhmann 1996).

Changes in the media environment, especially the widespread use of social media, pose questions about how the public sphere has changed in the last decade and how churches and religious denominations can find a way to be present in this new environment. It is important for us to see that the nature of mass communication is different from religious communication. This is why there is a constant danger that mass communication, even if it does nothing more than give news of religion and religious life with the greatest neutrality, by its own logic, can easily falsify them.

The following research questions were used to prepare this study:

How can religious messages be transmitted in a rapidly changing social space?

What are the new arenas of religious public life and to what extent are they suitable for the transmission of religious messages and the creation of religious dialogues in that context?

What is the impact of the internet on community participation in church life in terms of interactivity (dialogue) and participation (involvement)?

**Research hypothesis:**

Whether Facebook contributes to the functioning of a tradition-oriented community; whether online presence is able to complement and replace social communication; whether online presence strengthens the community.

The community uses the internet to operate ecclesiastical religiosity, so that, through online participation, the community is able to fully engage with the ecclesiastical institutional system.

## 2. Literature Review

Religiosity in reflexive modern societies is different from the religiosity of the past. Its main characteristic is mediality (Neville 2002). This is a new type of religiosity and the traditional church-based religiosity is in decline (Berger 1981; Derrida and Vattimo 1998; Huber 1998; Newbigin 1989). Nevertheless, we cannot say that religion and churches are not present in spaces of communication in reflexive modern society. The period of reflexive modernism is defined by electronic communication. Religion is widely present in public life; it has an impact on culture and politics (Neville 2002). The media environment of reflexive modernity is based on participation, observation, activism, and co-operation (Jenkins et al. 2013, 2015). These activities are important attributes of the communication of communities, which is why they provide a good framework.

Even if research on modernity and religion has long been forecasting that religion will lose its previous role in modernity (Kaufmann 1989), we can state now that this has not been the case. The role of religion in reflexive modernity, understanding religious phenomena, and research on religious experiences are issues that have received more emphasis in the social sciences and in research into religion. At the same time, changes in society do not leave religion untouched, which explains why its structure and characteristics are becoming more differentiated. In addition to the trends of decline, there are unexpected renewals and surges (Neville 2002; Derrida and Vattimo 1998; Bellah 1970; Geffré 2001).

The role of religion is constantly re-evaluated, which can be best detected in the marginalisation of traditional ecclesiastic religiosity. The discourse of the marginalisation of religion in modern societies is mainly elaborated in secularisation theory. Secularisation is described in very different ways by scholars (Taylor 2007; Martin 1978), also depending on which religion definition they work with and which aspect of societal change they place emphasis on. Secularisation models examine the role and status of religion within modern societies. Niklas Luhmann explains the shrinking role of religion by the fact that sub-systems are formed because of functional differentiation in societies, and religion becomes one of those. The scope of religion (and of the church) decreases, it has no impact on the other sub-systems, and opportunities for influencing other fields are becoming slight. Media are also a sub-system, but their significance and societal role is constantly growing (Luhmann 1977). In Luhmann's functionalist approach, he explains the change in religion not by the extinction of religion or by the alteration of its functions but by the pluralisation of its functions, which results in more individual options.

Most of the theories of modernisation suggest that, in the sphere of societal institutions, there is a growing self-righteousness that results in the weakening of religion's macro-social integrating and leading role, which will disappear in time. This statement, however, does not say anything about individual religiosity, as these theories look at the role of religion from the perspective society as a whole on a macro level. To nuance the picture of secularisation in Europe, we bring in the phenomena of euro-secularisation and pluralisation (Davie 2000; Berger 1981; Pickel 2008; Tomka and Zulehner 2000). If we focus on Christianity, we can see that, on many continents—not counting an inner shift in the sizes of denominations—it has been able to strengthen its position (Bastian 2019). In Western societies, the process of secularisation is strongly related to many social groups. The socio-caritative, marginal nature of religion is the only one that still has an overall importance for society, besides which it is considered an aesthetic addition to Western holidays. The majority rejects the universal, normative lessons; they find them invalid and, moreover, some people actively oppose them. The divine legitimacy of the church—which affects everything, not only the affairs of the church—is already almost impossible to understand; it is not considered sustainable; furthermore, most people see it as an organisational trait of the church that makes them smile and which discredits the church in the modern world (Jitianu 2017).

In addition to secularisation, pluralism has an even stronger influence on modern societies; individuals of different faiths live alongside each other and their daily lives are influenced by different religious beliefs (Berger 1981, 2008). Globalisation has resulted in a



"worldwide dependence" that is showing signs of crisis such as markets without limits and widening division in society. This world has generated a "syncretism without profile", the laic state (which declares freedom of religion but refuses to be involved in or care about religion itself), the fundamentalist state (intolerance instead of tradition), and the state of technological materiality (profit as the benchmark for everything) (Jitianu 2017).

The threat of religions becoming invisible has been a topic of discussion since the 1960s (Luckmann 1967) and it has also been mentioned that believers have changed their links to religious institutions and a civil religion has formed (Bellah 1970) one based on a personal normative morality and values and builds, perhaps, on religious values but is no longer tied to a church. Competition has formed on the market of ideologies and values, where—for many reasons—traditional churches are at a disadvantage. This situation has led to personal religions and self-fulfilling strategies. The long ongoing and intensive search for personal religions echoes the problem identified by Johannes Thumfart, that is, the growing quantity of spiritual content present on the internet (Thumfart 2010).

After the first phase at the beginning of the appearance of the internet—when it had novelty value and there was more interest in louder content—the search for spirituality gradually moved onto this platform. The colourfulness of great variety is not a new and modern form of eradicating religion, as believed by many, but rather a new model (Casanova 1994; Dulles 1974). Nevertheless, it appears threatening enough in the eyes of a lot of people, a fear that is enhanced by the long-established aversion of religion to contemporary technology. The advantages and drawbacks of technology—with special regard to the dependencies they trigger—have been known for a long time (Cucci 2016). One in four internet users go online for some—widely understood—spiritual reason. The effects of this on religion and churches are not obvious yet, and it is not well researched or indeed at all. Notwithstanding that, we can say that it is a process of importance, even if we know that this interest is not directly on religious life or ecclesial organisations (Thumfart 2010).

There are some churches that already look at the online world as a natural space for their function. When it comes to the internal operation of a church (in internal communication), there are many methods for sharing and conveying relative information. This is not only true with regard to interpersonal communication among people belonging to that church but it is also true of group communication. The use of portals and websites as platforms for passive information sharing is quite common. The Catholic church throughout history has always used the available level of technology for its purposes; the Society of Jesus and other monastic communities were especially creative in this regard. However, in Catholic "public discourse", the appearance of new tools tends to appear as a threat because, this way, these notions could fit better into the general liberal–conservative, secular–sacral, etc., oppositions.

Today, many official ecclesial documents and speeches call for the rapid incorporation of new technologies into the "arsenal" of evangelisation and missionary work. "Mission" is in fact the magic word (Bosch 1991), one that has always been determining in the external communication of the church. The reason behind this is the fact that communication is, first and foremost, an effect on other people and, if this effect is of a religious nature, it is already a mission "par excellence". The most important moment and a paradigm shift in this regard was the Second Vatican Council, which marked the start of a new direction in ecclesial communication taking the changing times into account (Spadaro 2014). Likewise, the documents published by the Pontifical Council for Social Communications or the papal addresses delivered on World Communications Day are all important "meta-messages". They are clear points of reference that can guide ecclesial communication, be they oral—on the one hand, traditional (e.g., priest homilies) or, on the other, nowadays of laic origin—or written. This type of media use by the church fits into the paradigm: media as practice (Hoover 2006).

The technological in nature characteristics of mass communication—particularly of the internet—cause unique problems in ecclesial communication. Means of communication

can be classified based on how much they inspire individual reflection and how much they free individual imagination (McLuhan 1964). For the conscious representation of the written word and the unwritten, the unsaid must be imagined. Media prompt excitement by publishing *sensations*. If such sensations do not present themselves spontaneously, it is the media itself that creates "mass communication events". As a matter of fact, this logic is present in religion as well; this is the reason for marginal pieces of information relative to the Pope's travels (e.g., what kind of car he rides in) or extremities within the church (e.g., the priest who "sprayed" holy water from a car during the coronavirus pandemic). Many of these news items only became newsworthy because the media needed something to report that day (Dayan and Katz 1992).

The nature of mass communication is not the same as the nature of religious communication. That is why there is a constant danger that mass communication, even by simply giving news of religion and religious life with the greatest neutrality, still, in its own way, can easily falsify religion. It makes religion the object and the field of conflicts, fights, and sensations, which is the opposite of the explicit objectives and identity of most churches (Luhmann 1996). All this does not mean that, if bad things or sins appear in the church, they could not be referred to by external media—in the lack of options in internal communication.

Looking at the spread of the internet, we can see that its dynamic has surpassed that of radio and television. The two classical media have "made the way" for the computer network, and society has also proved to be more receptive: the opportunity to actively share information meant the end of the previous passive positions (McQuail [1977] 2006; Rogers and Dearing 1988). The internet is of a very complex nature, which is one of the difficulties in its study. It can be understood and studied as a technological tool, more precisely, as a technological environment, and as a structured system of computers and other digital gadgets. From another point of view, we can see it as an active agent which interprets communication content, as an active participant in the process of communication. There is a more popular opinion according to which it has already become a determining cultural space that is the principal scene and carrier of the most important cultural phenomena of our day; what is more, metaphorically, it can also be considered as an independent organism (Ryan 2010; Wallace 1999).

The internet does not only have advantages for communities that exist in virtual space. Given that the cost of online communication does not depend on geographic distance, it provides a cheap and effective solution for organisational and communicational support for communities; consequently, it can help strengthening relations between members of extant communities. Online communication is in real time and it can convey spontaneous emotional reactions (e.g., chat) but, at the same time, emailing systems and forums make communication possible among parties that cannot be in each other's physical presence. In addition to communication between individuals (one-to-one), there can be communication among the members of a community (many-to-many) or there is the opportunity to address masses (one-to-many). The latter, if conducted by offline media (e.g., the printed press), costs a lot; therefore, they are only available for smaller communities in a limited manner. In addition to an online presence, an offline presence and action are also important (Hoover 2006).

As a result of the constant flow of goods, persons, capital, knowledge, and cultural ideals in the space of globalisation, the seclusion of those communities that used to be entirely local seems to have ended. In societies made up of metropolitan groups formed along common interests, locally based organisations are difficult to interpret (Welmann 1999). Initial research often compared online communities to "real communities" and differentiated them; they tried to prove that these organisational units cannot be considered real communities, as they are unable to fulfil the most basic role and relevance of a community (Depew and Peters 2000). The important difference lies in communication within the community. The online space means that communication for some groups can only take place with the use of technology; therefore, even personal communication happens

publicly (for example, in the comments below a Facebook post). Since there are no time or space limits, everything happens "here and now"; instead of building a hierarchy, they tend to construct networks. Belonging to online communities is based on a free choice; members belong to these personal community networks not because of physical proximity but because of shared areas of interest.

The importance of online communication is obvious in the case of online communities (this gives room for their existence); nevertheless, it provides various opportunities for offline communities as well. Considering that its price does not depend on geographic distance, it provides an effective and cheap tool for communication among members separated by distance. In addition to strengthening those relations that already existed, it can help form and maintain new ones as well. Basically, it is better adapted to supporting the communication of local communities than any other medium before. It can be used for real-time communication (chat) or communication that is separated in time (e-mail and forums). While the communication among the members of the former group is moved into the online space, people who are unable to find an appropriate appointment time for real-time communication can keep in contact with the help of e-mails. If a traditional community "decides" to be present online as well, wittingly or otherwise, they become visible to users who do not belong to the community. Already, the fact that they have a website or a page on social media conveys information about the community in question, even in the case of a closed group that was only created for members of a particular online group, because, this way, even people who do not belong to this group learn about its existence. If the online activity of the community is public (for example, on social media), obviously a lot of information can get to non-members about the community—but, at the same time, these are pieces of information for members as well. Two factors influence the choice of a given platform when it comes to any community: one is technology; the other is the structure of social relationships (Campbell 2010).

Social platforms in the reflexive modern media environment have become inevitable, Facebook in particular, and we can consider them to be a new means of social communication. Social media were able to become so successful and could appeal to so many people because they builds on and strengthen basic human needs. In January 2023, there were 5.6 billion internet users worldwide and there were 4.76 billion social media users (Statista 2023). Mark Zuckerberg, on the 10[th] anniversary (Zuckerberg 2014) of Facebook, emphasised the important role the platform plays in online community building and he also set out the goal for the next couple of years, as he said they were going to be about sharing questions and complex problems. Lately, it has come to light that profit making overthrew this noble goal. Just as in the case of the other means of mass communication, when using the internet or Facebook, we should be aware of the basic characteristics of these tools. Similarly to the case of any other asset, there are arguments for and against it. An important attribute of social media is that they do not only open but seclude as well, because content is only available for those who are members of a group (De Querol 2016). The space of online communities directs people that belong there, the worldview of the followed group, activities and preferences on the page filter, and influence users. Online social platforms play a revolutionary role in collective co-operation (Rheingold 2005); in addition, they show a good example of co-operation without co-ordination (Shirky 2012). Arguments against have also multiplied recently; according to sceptical scholars, online community scenes can only limitedly assure co-operation and democratic functioning (Papacharissi 2010); moreover, they can become platforms for propaganda and the spreading of misinformation (Morozov 2011).

According to global data on Datareportal from 2021, 45% of young people between 14 and 24 use social media to learn about up-to-date news and current affairs, which, at the same time, means that, among respondents in this age group, social media are the most frequently used source of information (Datareportal 2023). Religious communities that use Facebook can have both open and restricted access; they can thus be part of the external and internal communication of the community (organisation) in question. Facebook, with

all of its positive and negative traits, can be used to get to crowds of people to provide information and it can have an important role in mobilisation. This is what churches and ecclesial communities should build on. Social media have had an important role in making personal religious beliefs visible and public. From a communications theory point of view, it is important to note that social media integrate scenes of interpersonal, group, or mass communication; several activities can be parallelly run on it. The convergence of levels of communication is a basic attribute of social media; it is important to be aware of that (Walther et al. 1972).

## 3. Methodology

### 3.1. Method of Data Collection

We applied quantitative and qualitative text analysis methods during our research. We studied the private Facebook page of the chosen group. We examined content posted in 2023 in the quantitative part of the research and we analysed posts from between 2021 and 2023 for qualitative data. We had asked for the permission of the moderator of the group, as we were only able to see these posts by becoming members of the group. We assessed the media use of members of this Facebook group in our quantitative research by counting posts and the responses they received (reactions and comments). As a first step, we marked the period we wanted to research. In this analysis, this period fell between 1 January 2023 and 8 June 2023, 159 days in total. Within this timeframe, members published 93 posts, the first on 1 January 2023 and the last on 5 June 2023. Based on data, we can say that 0.59 posts were published daily, which means that new content appeared on the page three days out of five.

During the collection of posts, we registered the following data: number, year, member who posted (by initials), the text of the post, the type of post, the number of reactions, the number of comments, and the number of members who saw the post. We registered the data in an Excel file, where we then also made further descriptive statistical calculations. The texts of the posts were copied word for word in every case; if it was not content with text, we inserted the photo into the file (sometimes the photo was of a text). We placed the posts into the following categories: text, text (foreign language), text + photo, text + photo gallery, text + link, text + selfie, photo, photo gallery, reference (article), and reference (music). Reactions were registered separately according to reaction buttons (like, love, care, haha, wow, sad, and angry) and we also kept count of comments and members who saw the posts.[1] To analyse the media use of the group members, we looked at the posts, reactions, and comments, in addition to the number of people by whom the post had been seen.

For qualitative research, we examined every post published during the three years and we tried to identify correlating thematic lines. The tendencies characterising the online activity of this community were accurately outlined by looking at these three years. Within the qualitative analysis, we focused on highlighted texts and chats in detail and we looked for important elements in the internal communication of this community. Facebook categories helped our research as they provided a categorisation that marked the relevance and the quality of the pieces of content placed in our system. We examined posts, their topic, and the comments alongside a timeline, which we then turned into an extract. The extracts were then the basis of our category system, which outlined the different thematic lines.

### 3.2. Strategy of Data Analysis

For quantitative research, we created categories of data collection and analysis based on the uses (and gratification) model introduced by McQuail–Blumler–Brow. Uses in their model are learning and information seeking, personal identity, social relations, and diversion (McQuail et al. 1972). For this purpose, we categorised posts according to content as follows:

- Memory: about events, information released post-meetings, mainly photos;
- Fun fact: information not relating to the group's function, directs to outside source;
- Prayer: shared prayers, devotions;

- Information: information mainly on events, on event organisation (if there is no need for a reaction or an answer), introductions of new members;
- Asking for information: concerning event recommendations, signing up for events or information requested for organising events;
- Felicitation: there is the word "whish" or another word meaning the same (I think of you with love) in the post;
- Appreciation: there is the term thank you in the post;
- Event recommendation: prayers together, meetings, information about events, it often appears together with asking for information;
- Personal information: information on personal state, emotions;
- Religious thought: non-prayer religious content, it can be thoughts shared by external people.

After having created the categories, we matched them to uses (Table 1).

**Table 1.** Post categories matched to McQuail–Blumler–Brow uses.

| Use | Post Category |
| --- | --- |
| learning and information seeking | information |
| personal identity | Prayer religious thought |
| social contact, interpersonal relations | asking for information event recommendation personal information memory felicitation appreciation |
| diversion | fun fact |

The category of "information" obviously belongs to "learning and information seeking". *Prayer* and *religious thought* were perhaps the most difficult to pair with a use. The reason for creating this Facebook group was to organise an ecumenical pilgrimage and to inform applicants. The annual organisation of the pilgrimage and other religious events is still a central element of the group's online activity, Christian thought and the experience and expression thereof have an important role in the lives of the members of the group—therefore, in the activity of the group as well. Consequently, every piece of content concerning religion (*prayer and religious thought*) is paired with the use "personal identity". Considering that, in addition to religion and the expression and experience of religious identity, event organisation and the maintenance of interpersonal relationships are also highly important in the group's activity, posts about events (*event recommendation and asking for information*) and personal content (*personal information, memory, appreciation, felicitation*) were paired with "social contact, interpersonal relations". Finally, we put content not strictly relating to the function of the Facebook group, which is usually from an external source (*fun fact*), into the use "diversion". After categorising posts and matching them with use groups, we tried to understand the purpose for which the members of the Facebook group "*Ökumenikus zarándoklat csoport* [Ecumenical pilgrimage group]"[2]—that, in real life, form a common religious community as well—use this social media platform.

We started qualitative research by identifying the thematic, based on the content previously gathered. During the research, we examined the posts along thematic lines by taking into consideration the objectives of these content lines, how they contribute to the interactivity of community communication (dialogue), and how much they facilitate participation, that is, participation in ecclesial life. When using social media, we even paid attention to what the community uses the internet for, that is, how and in what measure the posts and content shared represent the ecclesial commitment of the community, and, consequently, their belonging to ecclesial structures. After gathering the data, we formed

categories which were more or less the same throughout the three years. Detailed content was then matched with the categories that outlined the most important thematic lines used by the community. The structure of Facebook thematic lines and their implications meant they could only be examined and analysed retrospectively. This was mostly related to the events of the religious calendar.

## 4. Research Results

### 4.1. The Community in Focus: Ecumenical Pilgrimage Group

The Facebook group "Ecumenical pilgrimage group" was created on 11 December 2019, under the name "Ecumenical pilgrimage 2020". It can be seen from the original name of the group that it was created for the participants of a pilgrimage planned for the following year by its administrators, supposedly to facilitate organisation and communication. The group's name was changed on 24 August 2020 to its current one, and its function also changed; instead of the support of one event (pilgrimage), it became a social platform for the community and was active all year round. In 2023, there are 61 members in this group, two of which are administrators. The group is a private and secret one; therefore, it is only accessible for its members (for non-members, it is invisible on Facebook). Administrators have to consent to welcoming new members after invitation; this way, every new member is a personal acquaintance of at least one current member.

The spirit of the pilgrimage and, therefore, of the group is well represented by the fact that there is a monk, a priest, a Calvinist, and a Lutheran pastor among its members and spiritual leaders. The group is a temporary community. Despite there being a constant core of members, the participants on every annual pilgrimage vary; this is why we cannot consider it to be a permanent community. The Facebook presence of the members and the existence of the group show that there has been a demand for permanence, which could be perfectly provided by this social media platform. The substance of the ecumenical pilgrimage group is the annually organised pilgrimage on foot. The pilgrimage has its own website[3] that, on the one hand, contains all the information relative to its organisation and, on the other hand, it is a sort of archive for content from preceding years. The group has a constant team of organisers (eight people), who are also present on the platform. They are the group's most active members and organise events such as common prayer and one-day events. The latter they have been doing since February 2023. The group has one moderator.

### 4.2. Results of the Quantitative Analysis

During the timeframe in the focus of the research, 93 posts were published by the members. Most of them contained text (85 posts) and the majority of the posts (52 posts) belong to the category "text + photo". Only 5.39% of posts (five posts) contained some external link (e.g., to an article or video), which means that 95% were either own content or external content without a link to the source. This can be relevant because, in this way, to view the post content, members do not have to "leave" the group, they stay in it, so, to use reaction buttons or to write a comment, they do not have to return to the group platform (or to Facebook).

If we look at the number of posts according to their categorisation based on content, we can see the following: almost 30%, 27 posts altogether, belong to prayer; this is also the biggest category in the researched timeframe. We have to note, however, that, among other festivals, Lent and Easter (Lent: 22 February–8 April 2023; Easter: 9–10 April 2023) fell in this period. These are especially important to Christians, which could explain the high number of posts belonging to the category of prayer. The second biggest category is religious thought; altogether, 17 (18.28%) such posts were published by members in the researched period (this number was also influenced by the holidays mentioned above). The category of memory also amounted to more than 10 posts (12; 12.9%), but every other category had fewer than 10 posts in the timeframe we looked at. It is important to note that, in the case of three posts, we could not decide whether they belonged to the category of event recommendation or asking for information, because both kinds of content had the

same weight in the posts; therefore, these were put in the category asking for information + event recommendation. This did not affect the analysis based on uses, because both categories belong to the use of "social contact, interpersonal relations".

If we summarise all the categories and the number of posts in all four uses, we obtain the following results. A remarkably high number of posts can be put into the uses "personal identity" and "social contact, interpersonal relation". While a total of 39 posts belong to the latter (41.94% of all the posts), the majority of the posts can be found in "personal identity": 44 posts, which is 47.31% of all the content posted during the timeframe in focus. If we add these two uses, almost 90% of posts belong to these two groups. In the other two use groups (learning, information seeking and diversion), there are five posts, respectively.

Group activity in the researched period can be described as follows: on average, every post was seen by more than two thirds of the members (68.17%), 20.57% of whom used one of the reaction buttons—which meant almost 13 reactions to each post (12.49). An average of four comments arrived to each post (4.23) but, given the fact that a member—unlike in the case of reaction buttons—can write more than one comment on the same post, these data cannot be used to calculate what percentage of members on average commented on posts in the researched timeframe (only the number of comments was analysed; the number of commentors was not).

The primary objective of our research was to assess how members of the Facebook group "Ecumenical pilgrimage group" use media; for this, we rated the reactions (reaction buttons and comments) to posts published during the researched timeframe, and we examined which category or use group we can put them in. We created rankings based on the use of reaction buttons, comments, and all reactions (reaction buttons + comments). In every ranking, we looked at the upper 10% in detail, that is, the nine posts that triggered the greatest response. Find the rankings in Table 2.

**Table 2.** Post rankings based on the number of reaction buttons, comments, and all reactions (post categories, upper 10%).

| Rank | Use of Reaction Buttons | Comment | All Reactions |
|---|---|---|---|
| 1. | felicitation | event recommendation + asking for information | event recommendation + asking for information |
| 2. | memory | religious thought | religious thought |
| 3. | appreciation | event recommendation | event recommendation |
| 4. | religious thought | prayer | prayer |
| 5. | memory | prayer | prayer |
| 6. | prayer | prayer | prayer |
| 7. | religious thought | prayer | prayer |
| 8. | prayer | prayer | prayer |
| 9. | event recommendation | prayer | prayer |

If we rank the use group that posts belong to, we obtain the following results (Table 3).

As can be seen in Table 2, the ranking based on comments and the one based on all reactions are identical, and not only the categories but the comments themselves too. This does not appear to be coincidental, as all reactions are a sum of comments and reactions. It is probable that, if a user writes a comment on a post, they will also use one of the reaction buttons as well. Then, the reactions from users that did not comment are also added. We should bear this in mind when comparing the rankings based on only rection buttons and only comments. While, in the ranking based on reaction buttons, we can find six different categories in the first nine posts, this number reduces to four in the ranking based on comments and, among the nine posts in this ranking, six belong to the category of prayer.

The diversity of every ranking reduces when we look at them based on use groups. There, we can only find posts belonging to the use groups "personal identity" and "social contact, interpersonal relation". Even if these correlate with the fact that most posts are from these groups, this phenomenon is not necessarily the case. Those few posts (10 in total) that can be matched with the other two use groups could have triggered intense activity from users (e.g., there may have been many reactions to an interesting post); based on data, however, we can see that this was not the case. If we tighten the ranking even further and only look at the three posts that received the biggest response, the variety of use groups also diminishes further: while in the ranking based on comments, we can see "identity" in the first three rows, in the ranking based on reaction buttons, we can only find posts that belong to the group "social contact, interpersonal relation".

**Table 3.** Post rankings based on the number of reaction buttons, comments, and all reactions (use groups, upper 10%).

| Rank | Use of Reaction Buttons | Comment | All Reactions |
|:---:|:---:|:---:|:---:|
| 1. | social contact, interpersonal relations | social contact, interpersonal relations | social contact, interpersonal relations |
| 2. | social contact, interpersonal relations | identity | identity |
| 3. | social contact, interpersonal relations | social contact, interpersonal relations | social contact, interpersonal relations |
| 4. | identity | identity | identity |
| 5. | social contact, interpersonal relations | identity | identity |
| 6. | identity | identity | identity |
| 7. | identity | identity | identity |
| 8. | identity | identity | identity |
| 9. | social contact, interpersonal relations | identity | identity |

### 4.3. Results of the Qualitative Analysis

During the qualitative research, the following main thematic lines could be identified based on our research question: interactivity (dialogue); participation—in ecclesial life; spirituality; and community. Within the main categories, we could identify sub-categories, based on which we could examine thematic lines. The thematic lines of these two and a half years were identical in many instances; we only note the year itself if the thematic line in question was special and determining in some aspect.

### 4.3.1. Interactivity (Dialogue)

In this category, the thematic lines of relation building and opportunities to personally meet were the most determining. Posts on reading experiences also fell into this category, which allowed members to relate to one another (interactivity) by sharing experiences of faith. In between personal meetings, important pieces of news were shared thanks to this virtual platform. Posts related to the thematic line of common faith, as an alternative, posts about illness, medical examinations related to an illness, and important events in members' lives appeared. Prayers shared on the platform are an attribute of the religious life of the community itself, which played an important role in creating the group on the one hand, while contributing on the other to the maintenance and raison d'être of the group. Sharing prayers was an objective at the time of creating the group and, with the contribution of the moderator and some members, this kind of activity never ceased. Prayers belong to more than one subject: they follow the unfolding of the ecclesial year, and different prayer subjects appeared as is usual in religious practice. The thematic lines of the prayers

vary: request/appeal, asking for forgiveness, acceptance, asking for blessings, forbearance, appreciation, listening, devotion, and holy relation. Dialogue is not only among members of the group but dialogue with God through prayers also receives emphasis. Dialogue with God is shaped by the prayers, and, in addition to prayers, there are citations from the Bible which further sophisticate the thematic line in question.

4.3.2. Participation in Ecclesial Life

Within this category, there are two different subject groups. One of them is the ecclesial year and its events and the elements of the thematic line thereof; the other is the monastic life of monks at the monastery of Bakonybél.

Thematic Lines Concerning Events of the Ecclesial Year[4]

Pentecost, the love of God—that lights people's souls (a photo was published of the coming of the Holy Spirit as illustration). There is also a comment on this post, a pilgrim interprets a confirmation in their community that it was the fire of the Holy Spirit that prompted 12 children to confirmation.

The thematic lines of the Holy week and of Easter concern seasonal events. During Lent, they concern preparations and physical fasting. Because of the nature of the holiday (celebrating resurrection), Jesus, his life, his sufferings, and death are at the centre. The thematic line of Palm Sunday is also about Jesus; here, the emphasis is on following and on the path to peace. At the beginning of Lent, on Ash Wednesday, the pastor moderator gave advice relative to spiritual attention and his post was more focused on this than the physical benefits of fasting, and he suggested the creation of an internal room that is suitable for personal retreat. Lent is an important period in the ecclesial year, so posting daily prayers (pilgrims) and weekly homilies (pastors and priests) gave the subtopics of the preparation for Easter. A prayer thread (daily prayers) was posted in all three years of the researched timeframe. The year 2021 was a year defined by the pandemic; therefore, death is a general topic but thanksgiving, gratitude, and requests as well. Quite a few members of the group were ill in spring 2021; as a result, the sickness topic was present (suffering and healing). Churches were still closed down during Lent; an element of the thematic line is an analogy for the resurrection of Christ who was not in his tomb because he had been resurrected. Absence is not always negative. The message of Easter is that losing ourselves is not always a loss; hope can come from hopelessness.

There is always a high level of activity when the ecumenical week of prayer is held. The moderator/pastor's thematic line for the ecumenical group is "miracle", that is, despite their differences, its members are able to experience unity. Miracle also refers to the kingdom of God that is "inside every one of us and around us" and has an impact on the whole year. The function of the group relates to the model of the ecumenical working; other subjects further strengthen this. "The church grows not by proselytising, but by attraction" (Pope Benedict), "We should not try to convince other people, but we have to bear witness" (Pope Francis). As a result of the nature of the group, ecumenism is an accentuated thematic line. The point of ecumenism is the brotherhood in Jesus, unity of the church, reconciliation, and unification. The photo attached to the textual post is the badge of the pilgrimage[5]; its birth already has a great meaning to the group; the verbal and visual content strengthens the topic.

From New Year's Day (this for believers has a different meaning), there are posts of greeting touching on the role of a community, common prayers, thinking of each other, and the call to participate in a joint winter hike; in addition to these, there are posts on asking for blessings. In "well-wishes", there is a great emphasis on healing, hope, peace, and happiness—here, there is a photo of an earlier pilgrimage to add weight to the content. While topics on New Year's Day formed around the thematic line of requests and preparation, New Year's Eve was obviously about thanksgiving. Topics of the new year are strengthened by the shared Irish pilgrim's prayer (it is one of the daily prayers on the pilgrimage on foot). There is reference to a well-known Hungarian writer from

Transylvania, a citation from him, which is about the topics of resumption and closure, oblivion and remembrance, gratitude and thankfulness. Another Transylvanian greeting is also there in the post: let the new year bring good and take away the bad.

The topic of Christmas is again about the life and, more precisely, the birth of Christ. Jesus growing up is an analogy to the elevation of life by cherishing spirituality. A strong item of the topic relative to the miracle of birth is how God reaches down towards people. Advent's thematic line was mostly influenced by the subjects of illness and liberation. Illness is present throughout the year (because of the serious health issues of two members). The relative psalm (Psalm 45)—God is help, he stands by us in whatever difficulty, that is why trust in him is constant.

### The Benedictines of Bakonybél

Being connected to the Benedictines of Bakonybél is also an important topic; here, we can also find posts that strengthen the thematic line of belonging to the monastery. A shared video sheds light on the monks' ways of life within the monastery of Bakonybél and the thoughts of Thomas Merton—a photo of a monk cycling through a field, the monastery as a centre of culture. Another thread of thematic lines also connects to the monastery—music, as a channel for communication with God—which also links the two communities (Benedictines and pilgrims) by praising the role of music in ecclesial life.

### 4.3.3. Spirituality

Connection to God and experiences of God are at the heart of this thematic line. This appears as personal experiences but, many times, it is connected to scriptures. The thematic line is built up of the presence of God, thanksgiving, gratitude, and belonging to a community. The other thread of this category is connected to Jesus, who is the rabbi and master, to whom it is a joy and grace to belong. Elements are the keeping up and rediscovery of tradition, forgiveness, and following Christ. At the same time, there are topics in relation to the ecclesial calendar that still belong to this category too. The moderator suggested sharing certain parts of scripture, which brought several different topics onto the platform. He asked members to share their favourite passage from the Bible and to give a personal explanation, to bear witness. Within the main thematic line (biblical topics), the following topics appeared: faith in God, connection, reading of the Bible, connecting with pictures, conversion story, personal growth in God: failures and the joy in having children, comfort and reassurance in God, God is the God of growth, and the thematic line of the Little Prince: one sees clearly only with one's heart. In addition to faith and hope in God, the thematic line of challenge also appears, which relates to one of the daily themes of the pilgrimage from the year before and, despite the fact that we can consider it a separate thematic line, this relates very strongly to the thematic line of faith in God by the following posts: hope, experience, learning, determination, searching for reason, and relying on God. In this topic, fighting COVID-19 also appears. God leads and there is creation in the ruins (reopening of churches). The topic of COVID-19 brings in the comparison with the period following the Babylonian captivity, when fear and worries, illness, loss, and confinement were part of everyday life, but then captivity ended.

In addition to sharing prayers, the thematic line of sharing psalms was also very influential for a long period of time in the communication of the group. Shared psalms add further content lines in the communication of this community. These topics can be seen in the table below (Table 4) in chronological order.

### 4.3.4. Community

In the thematic line of the community, there are several different topics, one of which is the experience of community at the pilgrimage (temporary, with references to the pilgrimages of previous years); the other is the community experienced online that is permanently kept alive and the main element of this was determined by the thematic line of the prayer community.

**Table 4.** Psalm topics (authors' own edition).

| Psalm Number | Thread of Topic |
| --- | --- |
| Psalm 100 | Call for praise—thanksgiving |
| Psalm 125 | God protects his people—trust |
| Psalm 9 | the Lord is good |
| Psalm 24 | God saves the true |
| Psalm 145 | God's rulership—praise |
| Psalm 45 | Royal wedding song—praise |
| Psalm 8 | Praising the Creator—praise |
| Psalm 42 | A Levite's grievance in exile—desire to be with God |
| Psalm 84 | Pilgrims' song—happy are the ones who go on a pilgrimage |
| Psalm 33 | Praising providence—praise, thanksgiving |
| Psalm 23 | The Good Pastor—thanksgiving, being grateful |
| Psalm 139 | God is all-knowing and present everywhere—God knows us best |
| Psalm 103 | Love of God—blessing |
| Psalm 4 | Evening prayer—thanksgiving, being grateful |
| Psalm 91 | Under God's protection—trust |
| Psalm 102 | Prayer in difficulties—complaint, request |

Pilgrimage

The thematic line of the pilgrimage is governed by several citations and textual illustrations: "Traveler, your footprints are the only road, nothing else. Traveler, there is no road; you make your own path as you walk. As you walk, you make your own road, and when you look back you see the path you will never travel again." (Antonio Machado, trans. by Mary G Berg and Dennis Maloney.) There is another citation from apostle Jacob about the gifts of God (Jacob 1,17). The thematic line is strengthened by audiovisual support (youtube video) and photos (e.g., footprint of a pilgrim). Information relative to the organisation of the pilgrimage also belongs here, and there are photos from previous years posted as well. According to the topic of the community, community is the essence of the Earth, Jesus is always reflected in terms of community, and this is a missionary activity. Posts about different meetings are not only interesting because of the information they convey but they also strengthen the sense of belonging within the community. These posts have visual and audiovisual content as well. An interview with one of the organiser/pastors, where he talks about his understanding of the pilgrimage also relates to the pilgrim topic. Inner journey is added to this topic here.

Praying Community

The main thematic line here is communion and the experience of a real community, where requests can be articulated, members can express how grateful they are, and there is the opportunity to pray for others. The online platform (virtual community) provides the opportunity to have access to the community independently of space and time. Matthew 18, 20, which is often cited in ecclesial communities, appears here too: "... for where two or three gather in my name, there am I with them" (MT, 18, 20). The thematic line is stronger because prayer is church, a space where the community is together and they pray and sing together. Community is a spiritual home, which means the presence of God and the sharing of love and of burdens. In sickness, prayers give strength and endurance to those in pain.

### 5. Summary of the Analysis

We analysed the media consumption of the Facebook group "Ecumenical pilgrimage group" using quantitative and qualitative methods to gather a comprehensive and detailed overview of the social media (Facebook) use of a tradition-driven community. In the quantitative part, we applied McQail–Blumler–Brow use groups to examine members' activity in the group based on the number and type of reactions to posts. Based on content, we created 11 categories that we put the posts into; these were matched with one of the four use groups. Almost 90% of the 93 posts that we looked at fell into the use groups of "personal identity" and "social contact and interpersonal relations", but this did not immediately mean that these posts triggered the greatest response from the group. At the same time, rankings based on the number of reaction buttons and comments yielded the following result: in the upper 10% of both rankings fell the posts belonging to the above-mentioned two use groups, so most reactions and comments were to these posts. Based on this, we can say that, for members of the "Ecumenical pilgrimage group", expressing and experiencing personal identity and maintaining social contact and interpersonal relations are the priorities when using this Facebook group. This result is in line with the objective of the creators of the Facebook group: to provide an opportunity for year-round communication and communication for the community built around an ecumenical pilgrimage and the organisation of a variety of events throughout the year. The two media use groups appearing in relation to members of the group are in line with this and with the need to live and strengthen the Christian spirit of the community within themselves and their own religion. It also allows members to maintain the relationship with the people they met at the events between pilgrimages or to meet future fellow pilgrims before the pilgrimage.

Christian pastoral care can be found in the nurture, enhancement, and deepening of transcendence experiences; in addition, it is also important to link these experiences to the Christian tradition. The experience of meeting with God needs 10 elements: loving your fellow humans, the power of mystery, the power of listening, lifestyle rules, transition—the joy of renunciation, time—desire to keep time, silence, the presence of God, reading one's soul, and prayer (Wikström 2013). The presence of the above-mentioned 10 elements can be detected in the life of the observed online community. The identified four main categories connect well to the 10 elements of the experience of meeting with God. The thematic lines correlate in more than one way with the elements, but let us not forget that this virtual community is a continuation of a physically existing community. Nevertheless, daily contact and communication are provided in the virtual space. Therefore, the virtual community, in addition to facilitating daily communication (dialogue) also provides pillars for the maintenance of communication with God. The other important aspect in the analysis was participation in ecclesial life, which was justified more than once during the examination of the thematic lines: by following the events of the ecclesial year, interpretations thereof, attribution of meaning, connections, and relations to the Benedictine order. We demonstrate that the community is tradition-driven by the fact that it is based on a Christian tradition, the pilgrimage.

Both hypotheses have been confirmed: the community maintains dialogue through online presence between the presence meetings, thus strengthening community communication; the community lives an active religious life on Facebook, which is reflected in the activities related to the church year, prayer life, and spirituality.

### 6. Summary

In this paper, we looked at the online activity of a tradition-driven community on a social media platform. The theoretical context of the empirical research was provided by the research on the role of religion in the modern age and reflexive modernity, but we also wrote about the relationship between religion and the media, at least when it comes to internet consumption, which meant a new field and a challenge for churches. During the research, we tried to find an answer to the question of how the requirements of forming a community can be met in the online, technological space. How can and how much do

Christian communities want to meet the requirements of this platform? Do they resist, sabotage, or, on the contrary, embrace the new rules of this space and will they adapt to their everyday function in their everyday practice? The research results show that, when it comes to this particular community, they can create a real community, a communion in the communication of the church. All this works in line with the changes in society, as always. The community is prone to judgement and control of its environment. This judgment can be disputed but it can also be engaged with in a constructive and patient dialogue. In this way, the community can show its own values, and opinions, which can represent the special identity that accommodates their value system; it allows their strong faith to come to light in a wider social environment.

**Author Contributions:** Conceptualisation: M.K.K. and I.J.B.; Methodology: M.K.K. and A.V.H.; Investigation: M.K.K., I.J.B. and A.V.H.; Writing—original draft preparation: M.K.K. and I.J.B.; Writing—review and editing: M.K.K., I.J.B. and A.V.H.; visualisation: M.K.K. and A.V.H. All authors have read and agreed to the published version of the manuscript.

**Funding:** This research received no external funding.

**Conflicts of Interest:** The authors declare no conflict of interest.

## Notes

1. In Facebook groups with fewer than 250 members when a member sees a post the post is automatically tagged "Seen by"—in groups of this size anyone can see the names of those who have read the post in question. https://www.facebook.com/help/530628541788770/?helpref=hc_fnav (accessed on 10 July 2023).
2. Ecumenical pilgrimage group.
3. http://okumenikus-zarandoklat.hu/ (accessed on 10 August 2023).
4. In our research we moved backwards on the timeline, therefore these topics will also be shown retrospectively.
5. The basis of the badge is the 12 crucifixes that symbolise the way of the cross that were put into the monastery at Bakonybél during its renovation and ordination in 2020. The medallion was designed and cast by one of the monks, it is called the Gerald-cross. One interesting fact about the medallion is that the crucifix is in negative, therefore it symbolises the lack of the crucifix, so it is acceptable to the Calvinist members of the group.

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
