# Peer review of "Tradition-Driven Religiosity on the Internet"

_religions, doi:10.3390/rel14111430_

Round 1
Reviewer 1 Report
Comments and Suggestions for Authors
The research conducted for this paper is thorough and diligent. It is particularly impressive that the communications have been categorised into nine categories and assigned to four purposes. The description, compilation, and arrangement of these elements are crystal clear, innovative, and do not require any additional work. However, I find the prolonged discussion of the spiritual and theological significance of some of the Church's rituals, primarily Roman Catholic, in section 4.3.2 to be tedious. If the reader does not understand what Lent is, he is unlikely to read the article. However, this section could accommodate literal examples. They would provide a sense of the tone, content, and manner of the discussions.
The essay's weakness is that the limited and comprehensive research topic are not identified until page 9. The first half of the essay consists of a series of general statements about the situation of churches and believers in late modernity and the World Wide Web. The author does not elucidate certain concepts, such as late modernity, so it is unclear why he or she does not use postmodern or second modern. While the first chapter contains some gloomy references to the later research topic, the essay (2nd chapter) posted on the Internet does not. The second chapter is readable but lacks an internal narrative structure. It is difficult to orient oneself due to the apparent disorder. The literature cited is also fragmentary, with references to studies on online groups from 1999 and 2000, but lacking a number of canonical studies, such as Jürgen Habermas' critique. Consequently, the first two chapters must be significantly revised.
Table 4 in subsection 4.3.3 represents a novel contribution to the thesis that goes well beyond the current research. The internal Facebook community comments that were analysed were compared to King David's psalms, demonstrating the pervasiveness of vernacular culture. This chapter should be written as a separate study, as it is pertinently new and has not been studied in this manner in a Hungarian context.
The methodology is convincing, exhaustive, accurate, and pertinent to the topic at hand. However, I found the methodology description to be lengthy. As a researcher, I would like to know if 97 comments are sufficient to derive any conclusions. The English used in the study is very legible, crystal clear, and enjoyable.
In conclusion, the paper contains solid research, relies on analysis that has been conducted professionally, and draws pertinent conclusions. However, I suggest editing and revising at the specified locations.
Author Response
|
1. Summary |
|
|
|
Thank you very much for taking the time to review this manuscript. Please find the detailed responses below and the corresponding revisions/corrections highlighted/in track changes in the re-submitted files. Your opinion was very important to us, the redesign will make the study easier to understand and follow.
Thank you for your comment on the thoroughness of the research. We thank you for your comments on further research on the topic (comparing psalms with the influence of popular culture), which we consider to be an exciting and forward-looking project. The critical comments on the individual sections are useful to us and have been corrected and improved accordingly. Comments on the essay are gratefully received. We have corrected the term late modern to the reflexive modern/modern as suggested by the proofreader, in order to make it easier to identify the period. The methodological description and research findings have been abridged as requested by the proofreader and with a view to clarity and consistency.
2. Point-by-point response to Comments and Suggestions for Authors
The introduction concludes with the research questions and the hypothesis, thank you for drawing our attention to the importance of this. In the summary of the research, we have also included an evaluation of the hypotheses. The essay has not been revised because, in our opinion, the essay thoroughly and coherently explores and presents the scientific context of the topic. Its main points lead from the larger societal picture (the relationship between society and religion, religion and mass communication, online communication) through smaller societal subsystems to the narrower topic. In the presentation of the results of the research, the correction you suggested has been made and the presentation of thematic content for the church's rituals has been shortened. The literal examples have not been included in the research results because the study was already very long (due to the double methodology) and the inclusion of examples would not have fit within the given scope.
|
||
Reviewer 2 Report
Comments and Suggestions for Authors
The paper is a very mature, competent work, the author seems to be experienced. This is primarily evidenced by the extensive theoretical introduction, which has the character of an independent essay on the topic of religion and (digital) media. There, the author very clearly, logically and coherently presented the basic current lines of thought and research within this topic, showing knowledge of key authors and works to which he appropriately and correctly referred.
The research part is also excellently presented. First of all, the design of the research sample should be appreciated as very suitable for examining the given topic. The quantitative-qualitative method of content analysis and variable setting is also optimal.
However, the research part is also accompanied by several minor imperfections:
1. Explicitly formulated research questions and possibly hypotheses are missing. The author partially comments on them later (e.g. on p. 10 and 16), or refers to them (p. 11), but it is requested to clearly state them at the beginning of the article.
2. There is a lack of data on the sample used in the qualitative part of the research (number of posts in the sample).
3. I am not quite familiar with the term "narratives" used in the qualitative part of the research. I understand narratives as messages carrying evaluation, opinion, conviction. I see the given categories more like "thematic lines" or "content lines". I recommend the author to specify the definition of this category (if he wants to stick to the narrative, it requires a clarifying explanation).
In addition, I found several unclear or confusing places in the text:
1. p. 4: "There has never been such a great interest in spirituality". I find this claim too bold and unsubstantiated in a scientific text, although it is frequent in the common rhetoric of religious leaders. I recommend underlining or reformulating.
2. p. 10: “Every post was seen by more than two thirds of the members (68.17%).” This is a misleading statement because it is probably an average number. If we want to claim "every post", it will be better to show the minimum and maximum percentage, so that it can be seen that even the minimum is more than two thirds.
3. p. 11: "If we shorten the ranking even further and we only look at the three posts that received the biggest response variety of use groups further diminishes: while in the ranking based on comments we can see 'identity' in the first three rows, in the ranking based on reaction buttons we can only find posts that belong to the group 'social contact, interpersonal relation'.” According to my understanding, this statement is not valid, because the first three lines in both columns are identical (social contact, interpersonal relationship). Maybe I missed something...
4. There are typos on p. 1 (fifth line from the bottom) and on p. 2 (eleventh line from the top).
Finally, it should be added that the topic is very current and this paper has a high potential of scientific and social contribution. I rate it as very suitable for Religions journal.
Author Response
- Summary
Thank you very much for taking the time to review this manuscript. Please find the detailed responses below and the corresponding revisions/corrections highlighted/in track changes in the re-submitted files. Your opinion was very important to us, the redesign will make the study easier to understand and follow.
Thank you for your kind editorial opinion. Thank you for the effort you have put into your proofreading. Your comments and opinions have helped us to improve our study and correct errors.
- Point-by-point response to Comments and Suggestions for Authors
The introduction concludes with the research questions and the hypothesis, thank you for drawing our attention to the importance of this. In the summary of the research, we have also included an evaluation of the hypotheses.
The sentence in the content of the essay has been deleted, thank you for pointing this out. ("There has never been such a great interest in spirituality")
For the qualitative research, we did not pay attention to the number of entries, as the quantitative analysis was carried out in the quantitative part. In the qualitative part, we focused on the topics and the discourses, opinions and activity around the topics.
We thank you for your suggestions for improvements to the narrative. We agree that in this case the term narrative may mislead the reader and is a much better indication of the content than your suggested terms thematic line, topic line or topic. This has been corrected accordingly.
The confusing statement found on page 10 of Chapter 2 has been corrected. Table 3 has been replaced, thank you for your clarifying comments. And thanks for the comments on typos and errors, these have also been corrected
Round 2
Reviewer 1 Report
Comments and Suggestions for Authors
This study examines the digital internal communication of a cohort of pilgrims across a span of two years. The examination starts with an extensive scholarly article that provides an overview of the status of the churches, spanning from Vatican II to the contemporary era. The primary emphasis is on the communication strategies employed by the churches.
The work has several notable strengths, including the implementation of an updated hypothesis system, a comprehensive and clear presentation of a vast body of literature, and the incorporation of three distinct categorization systems. Her analysis of the interpersonal exchanges inside the religious community demonstrates a noteworthy level of originality. One of the primary conclusions drawn from the research is that community privacy pertains to phenomena that are inherent to the collective functioning of the group, rather than focusing on individual personal problems. Hence, the compact community, regardless of its online confinement, surpasses the realm of privacy and does not only serve as an intermediary stage between the private and public spheres. The online, exclusive, closed group can be seen as an online manifestation of Jürgen Habermas' concept of the third public paradigm.
The author's demonstration of scientific maturity is evident in her abstention from critiquing the remarks. The author skillfully navigates the challenges associated with the topic by deliberately sidestepping the requirements of conformity, religious imitation, or any overblown and occasionally discredited manifestations of belief, as demonstrated in the provided illustrations.
It is unfortunate that the discussion on communication issues in the third paragraph of the introduction and the line of inquiry presented in the literature review have not been explored in a dedicated research study. The text has exhibited an enhanced level of Englishness, resulting in a better readability compared to its prior state. The article exhibits a combination of styles, characterised by an energetic tone in the initial section, accompanied by the use of heuristic assertions and sociological terminology. Within the realm of analysis, the satisfaction derived from the methodology surpasses the fundamental narrative of the research. The article is characterised by its extensive reading, pleasurable content, and notably refined style. The application of scholarship requirements is extensive.